# Perspective of Water-Use Programs in Agriculture in Guanajuato

Jesús Hernández-Ruiz [1] , Paula C. Isiordia-Lachica [1] , Ilse A. Huerta-Arredondo [1] , Ana M. Cruz-Avalos [1] ,
Arturo Ángel Hernández [2] , Ricardo A. Rodriguez-Carvajal [3,4] , Jorge E. Ruiz-Nieto [1]
and Ana I. Mireles-Arriaga [1,*]

1  Departamento de Agronomía, División Ciencias de la Vida, Universidad de Guanajuato,
   Km. 9 Carretera Irapuato-Silao, Ex Hacienda. El Copal, Irapuato 36500, Mexico;
   hernandez.jesus@ugto.mx (J.H.-R.); pc.isiordia@ugto.mx (P.C.I.-L.); ilse.huerta@ugto.mx (I.A.H.-A.);
   am.cruz@ugto.mx (A.M.C.-A.); jorge.ruiz@ugto.mx (J.E.R.-N.)
2  Instituto Nacional de Investigaciones Forestales, Agrícolas y Pecuarias (INIFAP), Campo Experimental La
   Posta, Km. 22.5 Carretera Federal Veracruz-Córdoba, Medellín 94277, Mexico; arturo.angelhdz@gmail.com
3  Departamento de Ingeniería Química, División de Ciencias Naturales y Exactas, Universidad de Guanajuato,
   Noria Alta S/N, C.P., Guanajuato 36050, Mexico; rodriguez.ricardo@ugto.mx
4  Dirección de la División de Investigación y Postgrado, de la Secretaría Académica, de la Universidad Virtual
   del Estado de Guanajuato, Hermenegildo Bustos Número 129 A SUR, Colonia Centro, Purísima del Rincón
   36400, Mexico; rirodriguezc@uveg.edu.mx
*  Correspondence: ana.mireles@ugto.mx

**Abstract:** Integrated agricultural water management (IAWM) encompasses multiple factors, necessitating the evaluation of performance across programs and involved entities and local consideration in different regions. This study analyzes the relation of allocation budgets and government agency programs in relation to the average annual availability of groundwater in Guanajuato State. Documentary investigation was conducted on public programs, aquifer availability, and agricultural land types over the period from 2017 to 2023. In the last six years, the amounts allocated to government programs from subsidies and donations have increased by 40%, leading to enhanced agricultural productivity in the state. Considering the agricultural types (rain-fed, irrigated, and protected) as separate variables, simple linear regression explains 97.8% of the variability in the DMA, indicating a decrease of 78.2 million $m^3$ and an increase in irrigated agriculture. The estimator for the budget allocated to public programs is $-2.21 \times 10^{-7}$, indicating that even if the resources allocated to government programs related to the use and exploitation of water in the agricultural sector increase, the DMA will continue to decrease. Regarding the agriculture area type, the estimator has a value of $-0.00237$, indicating that each rain-fed or irrigated agriculture unit established would result in an approximate reduction of 2370 $m^3$ of water in the DMA. Taking this into account, it is imperative to formulate strategies that consider intersectoral links, with a focus on prioritizing essential actions in rain-fed areas for water capture and/or irrigated agricultural areas for food production, which comprise 52% of the total land dedicated to the agricultural sector, and specifically targeting actions that promote groundwater management.

**Keywords:** water resource allocation; water budgetary programs; groundwater management availability





## 1. Introduction

The allocation of the general expenditure budget and the support programs promoted by the state government must be analyzed in relation to the average annual availability of groundwater (DMA) and the agricultural land area dedicated to production to ensure sustainable agricultural practices. Budgetary processes affect agricultural development, thereby influencing the growth of this sector and potentially leading to groundwater extraction rates that exceed the recharge rate, causing the diminishment of water tables and land subsidence [1,2]. This demonstrates the importance of a solid legal and institutional

approach for the sustainable management of groundwater resources, highlighting the relevance of effective policies and regulatory frameworks in budget allocation. Reports indicate that budgetary decisions and fiscal policies can influence not only economic growth but also the availability of water resources, which are key to agriculture [3,4].

Water for agriculture is a fundamental element for development, and it is the point on which the interests of users, resource managers, and social benefits should converge [5]. However, the excessive use of this resource raises the need to seek mechanisms that can be integrated into the development of programs and projects, highlighting the role of water as an environmental, social, and economic good [6].

Integrated agricultural water management (IAWM) must consider demographic, economic, technological, and environmental factors [7], which implies the necessity of knowledge and understanding concerning the performance of the entities and/or programs involved in water management and governance. When analyzing these concepts separately, it is first necessary to consider management as a technical capacity for hydraulic construction, including the appropriate infrastructure for maintenance and rehabilitation; then, labor and hydraulic governance must be considered as the processes of technical and social control responsible for its proper or improper use [8]. Implementing water reforms on a global scale is quite challenging, and it is particularly hard in Mexico. Former experiences have demonstrated how hard it is to transform policy intentions into measurable actions. Back in 1989, CONAGUA identified 'easy forms' to diagnose the challenges in water supply and management. Although many of the data consider 'common knowledge' as the greatest obstacle, aligning the incentives of various stakeholders has proven to be a significant impediment delaying the implementation process [9]. The responsibility for water resources lies with various agencies, as a shown in Figure 1, which can result in deficiencies in administration or intersectoral communication, leading to the uncoordinated development and management of water resources and producing social conflicts, waste, and unsustainable development [10]. These issues highlight the need for clear guidance on executing the reform roadmap outlined in the 2030 Water Agenda.

Water is scarce in Mexico's arid and semi-arid regions, where commercial agriculture is an important activity. These regions produce significant incomes from horticultural and fruits exportation as well as production for national consumption. On average, recharge is greater than withdrawals. However, reliance on groundwater has led to overexploitation of certain aquifers. Currently, at the national level, water systems pump about 29 km$^3$ of groundwater annually, with agriculture using over two-thirds of this total, resulting in aquifer depletion and, more often, irreversible water quality degradation [11].

In Mexico, there are great disparities between the different regions in terms of water quality, quantity, and availability. This is especially true in rural locations, which usually have low financial and human capacity to address their water needs, making it even more difficult to manage water resources. In contrast, in irrigation districts, where members are well identified and recognized as legitimate entities by the government, irrigation units typically operate based on informal arrangements, and they are neither monitored nor organized to communicate their needs. This lack of formalization restricts their participation in water management institutions such as COTAS (Technical Groundwater Committees) [9].

In the case of Guanajuato, the government has supported the establishment of 14 COTAS since 1998 as a complement to other measures to reduce groundwater extraction. This was a unique effort because the management board of a COTAS is composed exclusively of groundwater users. They coordinated its operational staff with the CEAG and receive technical support from CEAG staff, local universities, and technological centers [12]. Although the CEAG had no legal mandate regarding groundwater abstractions and management, in this approach, it felt compelled to support the COTAS to counter issues related to groundwater management [13].

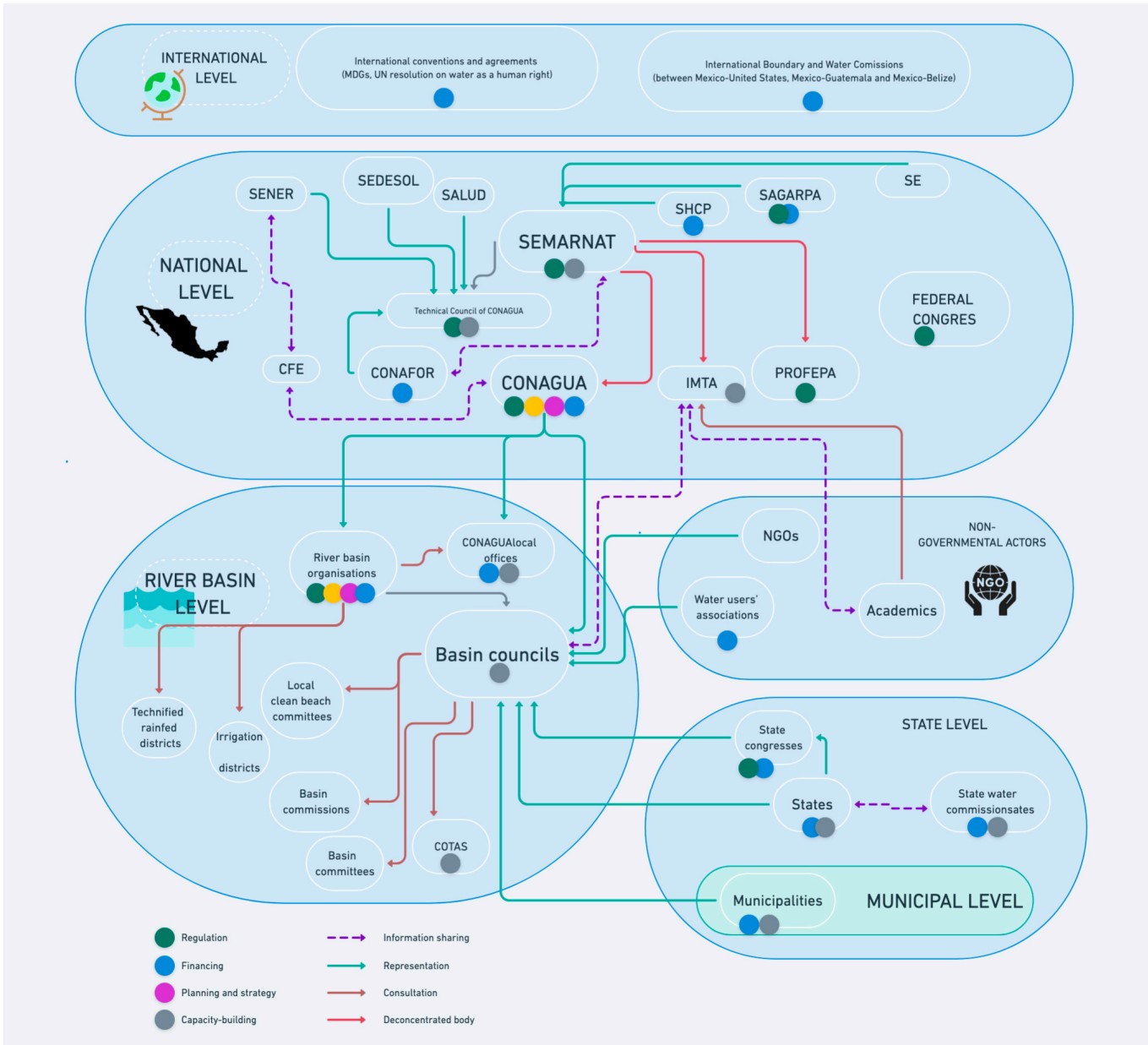

**Figure 1.** Institutional mapping of water resource management in Mexico: SAGARPA (Ministry of Agriculture, Livestock, Rural Development, Fishing, and Food Supply), CFE (Federal Electricity Commission), CONAGUA (National Water Commission), SE (Ministry of Economy), CONAFOR (National Forestry Commission), SEDESOL (Ministry of Social Development), COTAS (Technical committees for groundwater), SEMARNAT (Ministry of Environment and Natural Resources), IMTA (Mexican Technological Institute for Water), SENER (Ministry of Energy), PROFEPA (Environmental Protection Federal Attorney Office), and SHCP (Ministry of Finance and Public Credit) [9].

It is well known that there are significant socioeconomic differences between urban areas and rural regions in the state, especially in regions dedicated to livestock or agricultural activities. Moreover, there are huge differences between agricultural methods. Regulatory and participatory strategies to control the expansion of groundwater extraction had little success, especially in the southern region [14]. In 2024, there were 600 identified aquifers [15], which later reduced to 425, with 256 cataloged as groundwater in 2023 [16]. Although the precision of the groundwater balance is open to question, these aquifers have long been excessively abstracted, leading to a substantial annual overdraft on

aquifer storage [13]. Therefore, it is necessary to analyze the overall information to improve management decision-making, especially at the regional level.

The modernization and development of the agricultural irrigation in Guanajuato State occurred in two stages: one endogenous and one exogenous [17]. The first stage occurred during the 1960s and 1970s, marked by a transition from maize and beans to sorghum, used in balanced animal feed, especially for the poultry and pork industries. This transition was largely driven by governmental support in pricing, credit, research, and technology transfer, aiming to bolster agro-industrial complexes for meat and milk [18], principally under the dominance of transnational companies such as Ralston Purina, Anderson, and Clayton [12].

The second stage, characterized by external development, began in the 1980s with the consolidation of large-scale exports of frozen horticultural products. This included the establishment of Birds Eye for frozen production and Heinz, Campbell's Soup, and Del Monte for vegetable canning [19]. The demand for fresh fruits and vegetables resulted in the growth of 'non-traditional agro-exportation'. Between 1980 and 1998, the cultivated horticultural area increased from 10,000 to 70,000 hectares, with the export value rising from USD 10 to 170 million [20].

The stages of agricultural sector growth occurred without a solid, sustainable water usage strategy, resulting in overexploitation and forming the uncoordinated basis for the expansion of irrigated agricultural land [12] (p. 156). In the 1970s, the associated demographic pressure led to considerable stress on groundwater resources. In 2001, the Ministry of Agriculture, Livestock, Rural Development, Fisheries, and Food (SAGARPA) reported a total agricultural area of approximately 1.2 million hectares, of which 408,000 ha was irrigated using groundwater through the operation of 11,603 agricultural wells [21]. At present, around 17,000 wells are abstracting in the order of 4000 $Mm^3/a$, which is estimated to be about 1200 $Mm^3/a$ more than the renewable limit [12]. The expansion of the population and agricultural activities did not consider the carrying capacity of aquifers, leading to a water deficit of 41.8 to 47.4% from 1977 to 1994. Water levels dropped below 150 m in depth and continue to decrease by 1 to 3 m per year [17].

By 2018, the Guanajuato State Hydraulic Program indicated an increase in agricultural land from 1.2 to 1.48 million hectares, representing a 23% increase. In this context, the Guanajuato State Water Commission (CEAG) recorded a 13% increase in groundwater extraction [22] (p. 177) for agricultural use, rising from 2924 $Mm^3/year$ in 1999 to 3378 $Mm^3/year$ in 2017 [17], with volumes further increasing in 2024 (Figure 2).

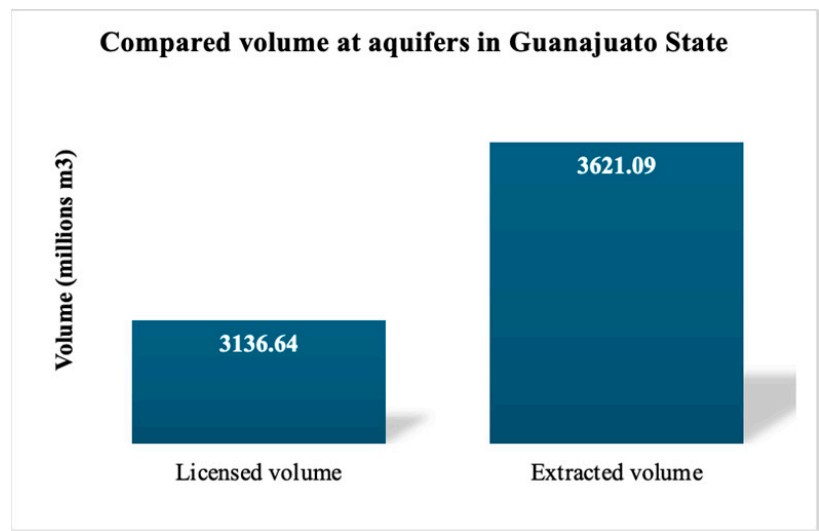

**Figure 2.** Volume comparison of aquifers of Guanajuato State in 2024 [22].

The intensification of water extraction, driven by the rapid expansion of groundwater irrigation, has transformed the economy, leading to significant growth in agricultural productivity and incomes. It is important to note that farmers' investments in pumping and irrigation equipment have simultaneously contributed to this expansion and decline water quality [14]. Analyzing groundwater management programs and budgets is a significant challenge. Despite the robustness and reliability of many studies [23–25], the precision and accuracy of equational models are restricted by their reliance on extensive datasets and grid sizes for differential equations. Such complex and sophisticated numerical modeling requires tools and knowledge that are not always available to all communities [26]. Although data on groundwater head time series are available in the literature [27], budgetary allocations [28] and considerations of different agricultural activity types at the regional level are not always included [27,29,30]. Therefore, the main objective of this research is to document the allocation of the state's general expenditure budget and support programs and their relation to the average annual availability of groundwater (DMA) and agricultural area designated for production in Guanajuato State from 2017 to 2023.

## 2. Review Methodology

### 2.1. Study Area

Guanajuato State is in the central–northern region of Mexico (Figure 3). It has a territorial area of 30,608 km$^2$, of which nearly 1.2 million hectares are dedicated to agriculture. By 2021, approximately 950,000 hectares were planted, with 416,690 hectares irrigated by both gravity-fed and groundwater-fed systems [29]. Groundwater extraction is primarily used for agriculture (76%), domestic use (21%), and industrial use (3%). According to CONAGUA [30], Guanajuato had 20 aquifers, 19 of which were overexploited.

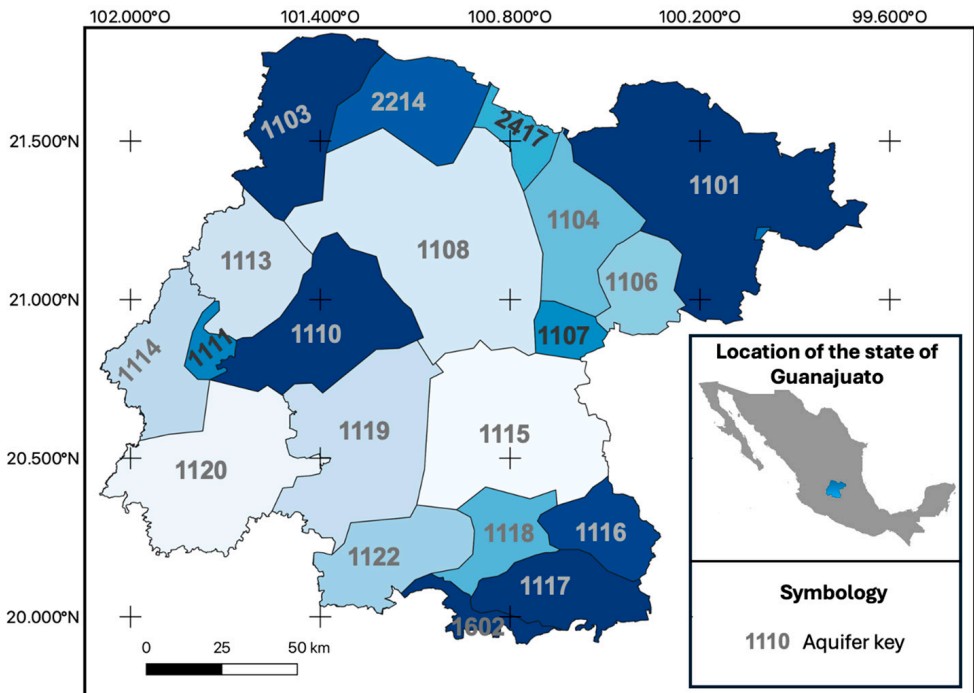

**Figure 3.** Geographic location of Guanajuato State and limitation of aquifers. The term "aquifer key" denotes the designation of the geographic boundary of the aquifer in question. The dark blue color signifies the average availability of the aquifer.

### 2.2. Information Search and Analysis

Documentary research was employed to collect, organize, and reflect on information considering three aspects:

(i) The characteristics of and amounts allocated to support programs related to water usage and utilization in the agricultural sector. This involved reviewing the amounts of budgetary programs in the Law of the General Budget of Expenditures of the Guanajuato State for the fiscal years 2017 to 2022. Additionally, official sources were consulted for the characteristics, objectives, and impact of the programs.

(ii) The value of the average annual groundwater availability (DMA) of the 18 aquifers located in Guanajuato State. The search was conducted in different official platforms, such as the CONAGUA database and reports issued through the *Official Gazette*, from 2017 to 2022 to observe the availability or deficit of the aquifers.

The DMA is the average annual volume of groundwater. When the value is positive, it indicates that the aquifer can be extracted for various uses, with consideration of the already concessioned extraction and committed natural discharge without endangering the balance of ecosystems. Conversely, a negative value indicates a deficit in the extraction [30].

(iii) Agricultural land area designated for production under irrigation conditions. Data were collected from the Statistical Yearbook of Agricultural Production (SIACON) for Guanajuato State for the last 10 years from 2013 to 2023 [31].

### 2.3. Quantitative Analysis of the Variables

Multiple linear regression analysis was conducted using Jamovi software (Jamovi, 2022; R Core Team, 2021) [32] to analyze the relationship between the dependent variable, i.e., the average annual availability of groundwater (DMA), and two or more independent variables, such as year, budget allocated, and agricultural land area (including area dedicated to irrigation, rain-fed agriculture, and protected agriculture).

Three multiple linear regression models were developed, considering year and budget allocated to public programs as independent variables, and each model considered a different type of agriculture as an additional independent variable. Model 1 considered irrigated agriculture as the independent variable, while Models 2 and 3 considered rain-fed agriculture area and protected agriculture area, respectively. This separation allowed for the observation of how each type of agriculture influences the DMA value.

## 3. Characteristics and Amounts of Budget Allocation of State Programs

The review of the general budget of expenditures of Guanajuato State for the fiscal years 2017 to 2021 showed that the budget allocated to the Ministry of Agricultural and Rural Development (SDAyR) averaged 2.7% of the total allocation for the public state administration, and that for the State Water Commission (CEA) was 2.1% of the total allocation of the administrative branch [33] (Table 1).

**Table 1.** Budget in millions of Mexican pesos (MXN) and percentage (%) assigned to SDAyR and CEA in the last six years.

| Year | Centralized Public Administration Budget (MXN) | Assigned to SDAyR (%) | Administrative Branch 30 (MXN) | Assigned to the CEA (%) |
|---|---|---|---|---|
| 2017 | 36,694.7 | 2.1 | 18,479.6 | 2.8 |
| 2018 | 37,783.5 | 3.1 | 20,752.3 | 2.3 |
| 2019 | 39,784.4 | 3.0 | 20,721.3 | 2.3 |
| 2020 | 38,771.8 | 3.1 | 19,361.5 | 2.2 |
| 2021 | 39,828.6 | 2.7 | 20,006.0 | 1.5 |
| 2022 | 38,125.1 | 1.7 | 20,993.1 | 1.4 |
| Average | 38,498.0 | 2.6 | 20,052,366 | 2.1 |

Source: Law of the General Expenditure Budget of Guanajuato State from 2017 to 2022.

It was observed that the budget for centralized public administration increased by MXN 1430.4 billion over the past six years, while the budget for the administrative branch increased by MXN 1526.4 [33]. However, the percentage allocated to the SDAyR and CEA decreased by 0.4 and 1.4%, respectively, with 2022 seeing the lowest budget allocation. This

observation is relevant because the SDAyR hosts the General Directorate of Agricultural Water Development (DGDAA), which coordinates actions to increase water productivity in agriculture and improve application efficiency through the installation of advanced irrigation systems. The CEA promotes water management strategies via hydraulic plans and programs, bringing water, drainage, and sanitation services closer to beneficiaries, but both government agencies observed a reduction in their allocated amount from the total budget.

Changes in public spending tend to focus on more visible components such as infrastructure or social programs. Public institutions and organizations do not always prioritize sectoral issues. In this regard, policymakers often prioritize certain organizations over others to attract different groups and enhance their public image [34]. These factors can influence the resources allocated to water management, including changes in fiscal policy and public spending allocation [15].

As shown in Table 2, the percentage allocated to programs addressing agricultural water use and management within the subsidies, grants, and social assistance expenditures budgeted for the SDAyR increased from 6.6% in 2017 to 18% in 2020. Subsequently, this allocated percentage saw a reduction in 2021 (15.6%) and 2022 (14.2%).

**Table 2.** Allocated percentage of subsidies, grants, and social assistance destined for the SDAyR for programs addressing the use and management of agricultural water.

| Year | Assigned to SDAyR (MXN) | TRA (%) | MRP (%) | MUA (%) | Total (%) |
|------|------------------------|---------|---------|---------|-----------|
| 2017 | 356.5 | 6.6 | - | - | 6.6 |
| 2018 | 422.8 | 11.9 | - | - | 11.9 |
| 2019 | 339.8 | - | 16.7 | - | 16.7 |
| 2020 | 446.9 | - | 16.7 | 1.3 | 18.0 |
| 2021 | 236.4 | - | 14.4 | 1.3 | 15.6 |
| 2022 | 283.9 | - | 12.5 | 1.8 | 14.2 |

TRA = Groundwater Irrigation Modernization; MRP = My Productive Irrigation; MUA = Better Water Uses in the Field. Source: Law of the General Budget of Expenditures of Guanajuato State from 2017 to 2022.

The analysis of the subsidies, grants, and social assistance allocated to the SDAyR in the general budget of expenditures for 2017 and 2018 shows that there was only one program, 'Groundwater Irrigation Modernization' (TRA), which was eventually transformed into 'My Productive Irrigation' (MRP) in 2019. According to the SDAyR [17], this is the same support program. The potential beneficiary population, as defined in Article 8 of the program's operating rules (2020), consists of agricultural production units with irrigated land. The target population, as specified in Article 9, includes approximately 40,000 agricultural production units with irrigated land using groundwater. The direct beneficiary population for the year 2020 was 1100 agricultural production units that met the access requirements.

In 2020, a new program, 'Better Water Uses (MUA)', was added. This program encouraged irrigated agriculture production units to adopt better practices through training. However, a report conducted by the SDAyR [17] identified that while the MUA program could be considered complementary to the MRP program, in practice, such complementarity does not exist. The MUA program addressed only a portion of the MRP beneficiary population, resulting in a lack of coordination between the two programs.

On the other hand, income from payroll and cedular taxes allocated to the SDAyR has resulted in an overall reduction of MXN 215,582 million. Despite this, programs addressing the use and management of agricultural water saw an increase of MXN 103,770 million. In 2022, the percentage allocated to these programs was 65.6%, representing a 40% increase over the past six years (Table 3).

**Table 3.** Percentage allocated from payroll and cedular taxes destined for the SDAyR, and percentage allocated to programs addressing the use and management of agricultural water.

| Year | Budget Assigned to the SDAyR (MXN) | MRP (%) | CA (%) | MUA (%) | ROH (%) | MTR (%) | CCL (%) | Total Assigned to Programs (%) |
|---|---|---|---|---|---|---|---|---|
| 2017 | 337.5 | 9.7 | - | - | 13.6 | - | - | 23.4 |
| 2018 | 42.3 | 14.2 | - | - | 11.6 | - | - | 25.8 |
| 2019 | 697.4 | 10.0 | 2.9 | 1.1 | 9.3 | 4.3 | - | 27.7 |
| 2020 | 636.3 | 14.1 | 5.0 | 0.9 | 4.8 | 4.7 | 0.1 | 29.6 |
| 2021 | 370.1 | 10.8 | 4.1 | 0.8 | 9.5 | - | 0.2 | 25.3 |
| 2022 | 122.0 | 32.8 | - | 4.1 | 28.7 | - | - | 65.6 |

MRP = My Productive Irrigation; CA = Let's Capture Water; MUA = Better Water Use in the Field; ROH = Rehabilitation of Surface Water Hydro-Agricultural Works; MTR = Comprehensive Modernization and Technification of the La Purísima Irrigation Module; CCL = Lerma-Chapala Basin Council. Source: Guanajuato State General Budget Expenditure Law from 2017 onwards.

The 'Let's Capture Water (CA)' program began its operations in 2019, aiming to increase the installed capacity for rainwater harvesting. Its main component is support for the implementation of constructed and/or rehabilitated border works. According to Article 4 of the operating rules of this program, its implementation and execution are carried out through the Borderia and Rural Infrastructure Trust for Guanajuato State (FIBIR), with agreements with federal, state, or municipal agencies or entities as well as with private institutions, universities, or any type of educational institution and organization. The target population for this program is 152,965 production units in Guanajuato State [35] (p. 40).

The 'Rehabilitation of Surface Water Hydro-Agricultural Works (RHO)' program aims to promote and support the development of necessary works and actions for the rehabilitation and modernization of hydraulic infrastructure in districts and irrigation units. These units are wholly or partially concessioned to beneficiaries to promote efficient irrigation water use [36]. This program is part of the Irrevocable Investment and Administration Trust fund for the execution of hydro-agricultural programs (FIDEA), created in 1997. Its operation involves a mix of federal–state resources and user contributions. In its first nine years of operation (1997 to 2006), it focused on the rehabilitation and modernization of the main "Ing. Antonio Coria" canal in 'Irrigation District 011 Alto Río Lerma', supplying low-pressure irrigation systems mainly in the Huanímaro and Corralejo irrigation modules, with approximately 18,641 hectares technologically enhanced (or technified as it is known in Mexico). Between 2006 and 2012, 40,953 hectares were technologically enhanced or 'technified' with low-pressure irrigation systems, primarily using PVC and multi-gate pipes. These activities are estimated to have saved an average of approximately 84.2 million cubic meters of water annually.

The 'Comprehensive Modernization and Technification Program of the La Purísima Irrigation Module (MTR)' finances projects and promotes coordination between CONAGUA and the state government with agricultural users. In 2011, a project was executed where pressurized irrigation systems were installed, leading to an estimated saving of 10 million cubic meters per year, which was allocated to the supply of the municipality of Irapuato [37] (p. 2).

The 'Lerma-Chapala Basin Council (CCL)' program had a budget for its implementation in 2019 and 2020, focusing on the proper distribution of water resources in the basin; however, in the program analysis [38] of the General Budget of Expenditures of Guanajuato State for the fiscal year 2020, no activities or indicators were found to provide further information on the development or impact of this program.

The General Budget of Expenditures of Guanajuato State for the fiscal year 2022 reported that there are 24 trust funds, with a balance of MXN 1,974,140,137.70 (one billion nine hundred seventy-four million one hundred forty thousand one hundred thirty-seven pesos) as of September 2021, distributed as follows: five belong to the Ministry of Environment and Territorial Ordering; five to the Ministry of Agricultural and Rural Development; three

to the Ministry of Education; three to the Ministry of Sustainable Economic Development; two to the Guanajuato State Water Commission; two to the Ministry of Government; and one each to the State Institute of Culture, the Guanajuato State Social Security Institute, the Ministry of Public Security, and the Ministry of Tourism.

In the last six years in the State of Guanajuato, six trust funds have opened (Table 4): the trust for social participation in water management in Guanajuato (FIPASMA), the Fund for Payment of Impacts for the Río Verde Project (FOPARIVER), the trust for the information system of the Lerma-Chapala Basin Agreement (Lerma Chapala), the Borderia and Rural Infrastructure Trust for Guanajuato State (FIBIR), the operational support trust for the Lerma Chapala Basin Council (FICUENCA), and the Irrevocable Investment and Administration Trust fund for the execution of hydro-agricultural programs (FIDEA).

**Table 4.** Annual resources from trust funds focused on programs that address the use and management of agricultural water.

| Year | FIPASMA | FOPARIVER | Lerma Chapala | FIBIR | FICUENCA | FIDEA |
|------|---------|-----------|---------------|-------|----------|-------|
| 2017 | 0 | 0 | - | - | - | 190,420,165 |
| 2018 | 114 | 268,500,003 | 818,639 | 0.54 | 4.27 | 77,060,322 |
| 2019 | 1,701,919 | 284,066,139 | - | 27,589,759 | 2,233,279.20 | 84,635,108 |
| 2020 | 1,142,060 | 292,267,601 | 917,906 | 33,322,547 | 1,350,161.07 | 178,006,545 |
| 2021 | 1,306,355 | 293,407,428 | - | 27,619,182 | 925,702.56 | 190,383,864 |
| 2022 | 1,007,405 | 299,075,093 | 0 | 9,497,991 | 1,519,980.54 | 176,066,065 |

Source: General Expenditure Budget Law of Guanajuato State from 2017 to 2022.

The FIPASMA trust was established with the support of the state government, initially planned to operate for up to five years (1999–2004). This allowed the Technical Groundwater Commissions (COTAS) to commence operations, with the expectation that they would become self-financing [22]. However, the trust continued to be considered in the General Budget of Expenditures of Guanajuato State for the fiscal year 2022. The proposed water management strategies do not address water overexploitation effectively, as they fail to consider the political dynamics of the country. These dynamics involve ongoing conflicts between political projects that influence and direct the organization of social units [39,40].

The activities designated for FICUENCA are detailed in Chapter III, Article 11 of the operating rules of this public investment and administration trust.

The FIDEA trust fund aims to construct, rehabilitate, complement, expand, and modernize the water capture, conveyance, and distribution infrastructure in Irrigation Districts and Rural Irrigation Development Units (URDERALES) to optimize hydro-agricultural infrastructure use. This is achieved through the allocation of cash resources to support and promote activities outlined in the technical annexes. Notably, financial information for this trust is publicly available electronically for the last three years (2020–2022).

The FIBIR fund focuses on financing projects to acquire machinery, equipment, and supplies for the execution of border works, rehabilitation and adaptation, flood control, rainwater catchment trough development, and pasture improvement. It also supports meeting infrastructure needs for producers in rural areas involved in agriculture, livestock, fishery, and forestry.

The main objective of a public trust fund is to act as a government tool for promoting development through investment and reinvestment in public works and social programs. In Guanajuato, Article 69 of the Organic Law of the State Executive Power states that public trust funds must be established exclusively to assist the State Governor in carrying out activities that are his own. Therefore, their use can be applied to practically all governmental areas.

In 2022, the Ministry of Finance, Investment, and Administration of the local government issued general guidelines for state public trust funds. These funds, as part of the parastatal public administration, aim to optimize public resource utilization for the benefit

of Guanajuato's population. Their purpose is to manage resources allocated to support specific programs and projects that assist the governor in fulfilling his duties.

However, Bulletin 6208 [41] (p. 6), which presents a reform initiative to the Organic Law of the Executive Power for Guanajuato State, indicates that public trust funds suffer from a lack of clarity and transparency in economic resources management. Historically, at the national and state levels, trust funds have been used by government entities and organizations to make use of large amounts of public resources without complying with transparency by oversight, surveillance, and control bodies.

## 4. Availability of Aquifers

According to a report from Guanajuato State's hydraulic program, 17 of 18 aquifers in the state are experiencing deficits, with only the Xichú-Atarjea aquifer showing availability. The report also notes that the deficit reported in the *Diario Oficial de la Federación* (DOF) is 484.45 million m$^3$ less than the figure calculated in the report. However, the year of the DOF issue used for comparison is not specified, nor is the method or value used to estimate the deficit for each aquifer in the state [22]. Official figures estimated the average annual groundwater deficit in 2023 to be −542.44 million cubic meters [42]. Similar to other arid and semi-arid regions, many water management problems, both in quantity and quality, are strongly linked to agricultural water use [43].

Information for the DMA of the aquifers in Guanajuato State was obtained from the agreements of the *Official Journal of the Federation* (DOF) published on 4 January 2018, 26 February 2019, and 9 November 2023 [42,44,45]. These agreements provide updates on the annual average availability of groundwater from the 653 aquifers of the United Mexican States, which are part of the Hydrological-Administrative Regions. Through these agreements, the National Water Commission (CONAGUA) announces the annual average availability of groundwater in the officially recognized aquifers in Mexico, based on the current Official Mexican Standard NOM-011-CONAGUA-2015 [33].

Over the past two years, the general state-level deficit has increased by 75.49 million cubic meters, with 13 aquifers identified as overexploited. The aquifers in the Celaya Valley, Upper Basin of the Laja River, Turbio River, and León Valley exhibited the highest values of deficit in the annual average availability of groundwater (Table 5). These data indicate a clear depletion of aquifers due to agricultural water use, which has become a point of pressure not only at the regional level but also globally, with significant implications for the management and sustainability of water resources. Reliance on groundwater for agricultural irrigation, driven by population growth, food demand, and climate variability, has led to alarming depletion rates of aquifers [46]. For instance, Alkon and coworkers have shown similar conditions in regions such as India, where groundwater levels are declining due to a combination of factors, including anthropogenic extraction, localized weather conditions, geological characteristics, and climate change [47]. This depletion threatens food security and agricultural productivity and affects the availability of water for drinking, industrial use, and ecosystem viability [48].

**Table 5.** History of the DMA of the aquifers of the State of Guanajuato over the last six years.

| Aquifer Name | Values in Million Cubic Meters per Year | | | |
| --- | --- | --- | --- | --- |
| | **2018** | **2019** | **2020–2021** | **2022–2023** |
| 1101 Xichú-Atarjea | 1.78 | 4.01 | 3.85 | 2.29 |
| 1103 Ocampo | −0.96 | 4.58 | 4.54 | 4.46 |
| 1104 Laguna seca | −31.17 | −29.85 | −31.84 | −28.02 |
| 1106 Dr. Mora-San José Iturbide | −23.36 | −23.30 | −27.01 | −36.71 |
| 1107 San Miguel de Allende | −9.99 | −9.51 | −9.9 | −12.4 |
| 1108 Cuenca Alta del Río Laja | −62.11 | −61.81 | −62.45 | −61.95 |
| 1110 Silao-Romita | 117.19 | 117.46 | 114.8 | 105.04 |

**Table 5.** *Cont.*

| Aquifer Name | Values in Million Cubic Meters per Year | | | |
|---|---|---|---|---|
| | 2018 | 2019 | 2020–2021 | 2022–2023 |
| 1111 La Muralla | −11.55 | −11.31 | −11.59 | −10.34 |
| 1113 Valle de León | −53.87 | −53.80 | −51.87 | −61.63 |
| 1114 Río Turbio | −52.92 | −52.32 | −53.35 | −54.27 |
| 1115 Valle de Celaya | −113.59 | −111.41 | −115.3 | −156.47 |
| 1116 Valle de La Cuevita | −0.48 | −0.42 | −0.06 | −1.23 |
| 1117 Valle de Acámbaro | 27.07 | 27.07 | 25.13 | 20.91 |
| 1118 Salvatierra-Acámbaro | −43.04 | −42.73 | −39.86 | −28 |
| 1119 Irapuato-Valle | −71.46 | −71.13 | −67.09 | −60.14 |
| 1120 Pénjamo-Abasolo | 127.89 | −127.40 | −128.2 | −126.11 |
| 1121 Lago de Cuitzeo | 2.77 | 2.77 | 2.72 | 1.53 |
| 1122 Ciénega Prieta-Moroleón | −11.02 | −10.25 | −19.54 | −39.52 |
| Total | −208.81 | −449.36 | −467.02 | −542.56 |

Own elaboration with data from [15,25–27].

## 5. Agricultural Surface

As observed in Table 6, agricultural areas in Guanajuato State have decreased. Specifically, the total cultivated agricultural area decreased from 1,074,541 ha in 2011 to 902,058 ha in 2023, a reduction of approximately 172,483 ha, which represents a 14% decrease in surface area. For irrigated agriculture area, this reduction was 63,305 ha. The expansion of protected agricultural areas significantly influences water consumption patterns, necessitating a shift to sustainable water management practices. To address this, it is essential to incorporate technologies that reduce water usage and improve crop productivity. For example, integrating rainwater harvesting systems in greenhouses can significantly increase the availability of irrigation water, facilitating crop productivity even during dry periods. The use of reclaimed water offers an alternative source that mitigates the overexploitation and contamination of aquifers in regions with high greenhouse gas emissions. Additionally, closed-loop greenhouse systems can further reduce water consumption [49–51].

**Table 6.** Total area and area planted under irrigation systems in the last 11 years in the State of Guanajuato.

| Year | Agricultural Area Planted in Hectares | | | |
|---|---|---|---|---|
| | Rain-Fed Agriculture | Irrigated Agriculture | Protected Agriculture * | Total |
| 2023 | 434,506 | 463,975 | 3578 | 902,058 |
| 2022 | 429,072 | 488,288 | 2919 | 920,279 |
| 2021 | 450,245 | 467,057 | 2681 | 919,983 |
| 2020 | 455,562 | 470,562 | 2739 | 928,862 |
| 2019 | 450,994 | 494,875 | 2372 | 948,240 |
| 2018 | 465,058 | 487,155 | 1883 | 954,096 |
| 2017 | 456,724 | 478,909 | 1672 | 937,306 |
| 2016 | 466,839 | 454,731 | 1047 | 922,617 |
| 2015 | 506,214 | 478,783 | 1177 | 986,174 |
| 2014 | 545,178 | 475,817 | 971 | 1,021,966 |
| 2013 | 569,128 | 477,429 | 807 | 1,047,364 |
| 2012 | 549,166 | 496,569 | 717 | 1,046,452 |
| 2011 | 541,497 | 532,608 | 437 | 1,074,542 |

Own preparation with data from the Statistical Yearbook of Agricultural Production [29]. * Surface under the greenhouse, shade mesh.

The Ministry of Agriculture and Rural Development reported that in the past ten years, agricultural production increased from 6,836,000 to 9,189,000 tons, translating to a 34% increase, and irrigated production rose from 6,108,000 to 8,061,000 tons, representing a 32% increase [52].

These figures indicate that, despite the reduction in the areas allocated to agricultural activities in the State of Guanajuato, productivity has risen. For example, maize yield increased from an average of 8.77 to 10.20 tons/ha [31]. Similarly, other crops also saw increases, such as the increase from 56.1 to 79.5 tons/ha in alfalfa (42%), from 3.1 to 7.7 tons/ha in asparagus (148%), and from 11.3 to 32.5 tons/ha in strawberries (186%) [52].

The increase in agricultural productivity in the Mexican agricultural sector is the result of incorporating improved varieties and seeds; optimizing plant nutrition, machinery, and infrastructure; and implementing technified irrigation systems [53–56]. This shift towards more advanced agricultural systems may involve some concerns about water use. Irrigation has been a subject of a significant debate regarding its impact on water resources. Some authors argue that intensive agricultural systems can lead to higher productivity while simultaneously safeguarding water resources [57,58]. However, other investigations point out that excessive use of external inputs in intensive agriculture can degrade soil and water quality [59,60] or promote the growth of products linked to internationally marketed agro-industrial value chains that do not contribute to national food security [61].

The fact that agricultural production relies on the intensive use of specialized inputs such as improved seeds, agrochemicals, and machinery fosters an exclusive modernization and leads to a duality in agricultural growth. There are producers who mainly own large extensions dedicated to production under irrigation systems, with a market structure influenced by transnational companies that concentrate on production and commercialization processes to efficiently supply the agro-industry [62,63]. In contrast, there are also small-scale agricultural producers who primarily practice rain-fed agriculture in areas with high levels of marginalization, with reduced production scales and no economic or technical capacity to access inputs, resulting in conditions of low productivity [47]. Despite the reduction in rain-fed land in 2022, representing 46.6% of the total land, it contributed to 54% of the total broccoli produced at the national level and maintained the production of sorghum and wheat. Notably, there has been a significant increase in agave plants from 91,669 in 2007 to 356,277 in 2022, showing an increase of 336%. This highlights the need for agave growth regulation in water management in the state [64].

There is evident development of inclusive policies to achieve equitable distribution among all field producers, granting them access to the productive resources they depend on to enhance productivity. Particularly, small-scale producers should be encouraged to conserve and sustainably utilize phylogenetic resources for food and agriculture [54].

Through this analysis, it is possible to identify the main determinants influencing agricultural productivity, including government expenditure, technological innovation, and international trade; it expresses that although more public spending is allocated, it does not lead to higher productivity in the sector [65], suggesting that productivity maintains its own facets beyond dependence on government incentives.

## 6. Multiple Linear Regression Models

The results of the multiple linear regression models, which predicted the average annual availability of groundwater (DMA) based on the variables of year; budget allocated to public programs related to the use and utilization of water in the agricultural sector; and the agricultural land area dedicated to irrigated, rain-fed, and protected agriculture (with a separate model for each of these last variables), are presented in Table 7.

**Table 7.** Coefficients of multiple linear regression models for the mean annual availability of groundwater (DMA).

| Predictor | Estimator | Standard Error | Statistical t | *p*-Value |
|---|---|---|---|---|
| Constant | 158,891.61 | 1239.71 | 128.20 | 0.005 |
| Year | −78.26 | 0.60 | −129.0 | 0.005 |
| Budget | $-2.21 \times 10^{-7}$ | $3.10 \times 10^{-9}$ | −71.10 | 0.009 |
| Irrigation | −0.00237 | $7.98 \times 10^{-5}$ | −29.70 | 0.021 |
| Model 1: DMA = 158,891.61 − 78.26 (Year) − $2.21 \times 10^{-7}$ (Budget) − 0.00237 (irrigated) $R^2$ = 0.978; $R^2$ adjusted = 0.95 | | | | |
| Constant | 82,745.51 | 1739.97 | 47.60 | 0.013 |
| Year | −42.054 | 0.842 | −50.00 | 0.013 |
| Budget | $-2.14 \times 10^{-7}$ | $2.24 \times 10^{-9}$ | −95.50 | 0.007 |
| Irrigation | 0.00413 | $1.01 \times 10^{-4}$ | 40.70 | 0.016 |
| Model 2: DMA = 82,745.51 − 42.054 (Year) − $2.14 \times 10^{-7}$ (Budget) + 0.00413 (rain-fed) $R^2$ = 0.985; $R^2$ adjusted = 0.970 | | | | |
| Constant | 159,342.25 | 189,582.13 | 0.84 | 0.555 |
| Year | −79.08 | 94.28 | −0.83 | 0.556 |
| Budget | $-2.17 \times 10^{-7}$ | $1.97 \times 10^{-7}$ | −1.10 | 0.469 |
| Irrigation | 0.0292 | 0.377 | 0.0774 | 0.951 |
| Model 3: DMA = 158,891.61 − 79.08 (Year) − $2.17 \times 10^{-7}$ (Budget) + 0.0292 (protected) $R^2$ = 0.956; $R^2$ adjusted = 0.824 | | | | |

The analysis of the multiple linear regression models revealed that Model 1 explains 97.8% of the variability in the DMA. The three predictors used are statistically significant at the 0.05 significance level. According to the estimators of this model, it can be inferred that for each passing year, the DMA decreases by 78.2 million cubic meters of water in the State of Guanajuato. The estimator for the budget allocated to public programs is $-2.21 \times 10^{-7}$, indicating that even if the resources allocated to government programs related to the use and exploitation of water in the agricultural sector increase, the DMA will continue to decrease in the aquifers of the State of Guanajuato. Regarding the irrigated agriculture area, the estimator has a value of −0.00237, indicating that each additional hectare of irrigated agriculture established would result in an approximate reduction of 2370 m$^3$ of water in the DMA.

The results are similar to other studies in different areas, where groundwater depletion and the sustainability of irrigation have caused significant changes in river systems, such as the transformation of perennial rivers into ephemeral rivers due to the pumping of groundwater for agriculture. This alteration in river flow patterns highlights the direct impact of irrigated agriculture on aquifers and surface water resources, emphasizing the interconnectedness of groundwater and surface water systems. Moreover, the conventional approach to irrigated agriculture, primarily focused on maximizing crop production, contributes to the widespread salinization of groundwater resources [66,67].

Model 2 explains 98.5% of the variability in the DMA. The three predictors used are statistically significant at the 0.05 significance level. According to the estimators of this model, it can be inferred that for each passing year, the DMA decreases by 42.05 million cubic meters of water in the State of Guanajuato. The estimator for the budget allocated to public programs is $-2.14 \times 10^{-7}$, indicating that even if the resources allocated to government programs related to the use and exploitation of water in the agricultural sector increase, the DMA will continue to decrease. In relation to the area of rain-fed agriculture, the estimator has a value of 0.00413, suggesting that each additional hectare of rain-fed agriculture established could increase the DMA by approximately 4130 m$^3$ of water.

Although this relationship is not strictly proportional to the establishment of rain-fed agriculture, it suggests that the application of certain strategies could favor the recharge of groundwater in the aquifers. Indeed, it is possible to identify key practices that can positively impact groundwater recharge within dryland agricultural environments. For

example, creating recharge basins and promoting soil conservation practices, such as the implementation of minimum or zero-tillage techniques, reduce erosion and favor water infiltration into the soil. Additionally, planting cover crops during fallow periods can help improve the soil structure and increase the water retention capacity, thus facilitating aquifer recharge [68–70].

In both Models 1 and 2, it is evident that the resources allocated to governmental programs for agricultural water use, without impacting aquifer recharge, pose a crucial challenge. This challenge involves generating an integrated and strategic approach, which must carefully consider the relationship between groundwater availability and the agricultural area dedicated to production. Adequate management of groundwater resources in relation to human activity and land use changes is essential to ensure the availability of water without compromising aquifer recharge.

Model 3 explains 82% of the variability in the DMA. The three predictors used are not statistically significant at the 0.05 significance level. These results suggest that there is no correlation of the variables of year, budget allocated to public programs, and protected agricultural area with the dependent variable of average annual groundwater availability (DMA).

## 7. Conclusions

Despite the decrease in the percentage of the budget allocated to the Ministry of Agrifood and Rural Development (SDAyR), the subsidies, grants, and social aid for programs related to agricultural water use and management have increased by 40% over the past six years. These resources are primarily allocated to programs that prioritize the implementation and maintenance of hydro-agricultural works and the modernization of irrigation systems, enhancing agricultural productivity. However, this increase in resources may be not reflected in groundwater conservation. Over the past four years, the average annual availability of groundwater has decreased, and 15 out of 18 aquifers are overexploited.

This study suggests including various approaches, such as integrated pest management, conservation agriculture practices, and integrated soil management, and different types of users, production units, and governmental stakeholders. These strategies aim to contribute to a better understanding of the overall problem. Additionally, the study recommends providing incentives for agricultural inputs and increasing financial support for programs such as the My Productive Irrigation (MRP) and Better Water Use in the Field (MUA) programs. It is crucial that the MRP program focuses on reducing the water deficit rather than merely generating operational rules. Given that the beneficiaries of water management belong to a different government levels, it is important to generate accessible data for policymakers and to coordinate allocation, spending, and investment at the regional level. This coordination should aim to avoid conflicts and duplications in responsibilities and projects in the near future.

**Author Contributions:** Conceptualization, J.H.-R.; writing—original draft preparation, P.C.I.-L., A.I.M.-A., and A.Á.H.; writing—review and editing, J.H.-R.; visualization, I.A.H.-A., A.M.C.-A., R.A.R.-C., and J.E.R.-N. All authors have read and agreed to the published version of the manuscript.

**Funding:** This research received no external funding.

**Institutional Review Board Statement:** Not applicable.

**Data Availability Statement:** Not applicable.

**Conflicts of Interest:** The authors declare no conflicts of interest.

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
