# Peer review of "Perspective of Water-Use Programs in Agriculture in Guanajuato"

_agriculture, doi:10.3390/agriculture14081258_

Round 1

Reviewer 1 Report

Comments and Suggestions for Authors

1.    It is necessary to add some quantitative description in the abstract

2.     The innovation of this paper needs to be highlighted in the abstract.

3.     Keywords need to be modified to reflect the research content of this paper

4.     The literature review is not enough, the innovation of this paper and the contribution made by previous studies have not been clearly expressed.

5.     Literature review and theoretical basis need to be described separately. The author should summarize the existing research gaps and highlight the innovation of this paper after completing the literature review. Theoretical basis should be the basis of model construction

6. The author needs to introduce the reasons for the research method and combine it with the innovation of the research in this paper.

7.  This article has obtained some interesting findings through the models, but these findings need to be further verified from theory or actual conditions. Also, further highlight the contribution of this article.

8.The article lacks an important discussion link, in which the author should focus on describing the differences between the article study and other scholars' studies, thus highlighting the relevance and academic value of the article.

Author Response

 Comment 1: It is necessary to add some quantitative description in the abstract

 Response 1: agree the quantitative data was added to abstract

Comment 2: The innovation of this paper needs to be highlighted in the abstract.

Response 2: agree the abstract was modified

comments 3: Keywords need to be modified to reflect the research content of this paper

Response 3: agree there are mofications to keywords

Comments 4: The literature review is not enough, the innovation of this paper and the contribution made by previous studies have not been clearly expressed.

Response 4:  Agree , there are specified data in the abstract

Comments 5: Literature review and theoretical basis need to be described separately. The author should summarize the existing research gaps and highlight the innovation of this paper after completing the literature review. Theoretical basis should be the basis of model construction

Response 5: agree, there are a separation about the documentary, data, and contruction models

Comments 6: The author needs to introduce the reasons for the research method and combine it with the innovation of the research in this paper.

Response 6: agree, the reason was added to a introduction section

Comments 7 This article has obtained some interesting findings through the models, but these findings need to be further verified from theory or actual conditions. Also, further highlight the contribution of this article.

Response 7: agree, some similar research was added the information was actualized to a 2023 the most actual budgetary data, nevertheless, for field research to study the actual problem, is necessary funding to that.

Comments 8: The article lacks an important discussion link, in which the author should focus on describing the differences between the article study and other scholars' studies, thus highlighting the relevance and academic value of the article.

Response 1: agree the conclusion was modified.

Reviewer 2 Report

Comments and Suggestions for Authors

This article presents significant findings that seem quite original. 

Overview of article:

The state or region produces for large-scale export, the framing of investment provides an objective assessment and recommendation against proven environmentally damaging practices, and almost urgent significant findings, with limited reason to expect a compromise to productivity with a reallocation of investment to recharging groundwater.

Despite an increase in percentage allocated to programs that address the use and management of agricultural water, the activities do not address most important almost urgent topics for water conservation, such as groundwater recharge which is an identified and significant trend in these study areas, with aquafiers being over exploited. It seems that importance given to irrigation systems, intensive productive systems on less land area, and rain fed small holder farming (clarification asked for) could be reason, amongst other reasons, stemming from an almost misallocation of investment funds.

While it mightn’t be the appropriate resource to recommend, your article could extend scope to international discussions about groundwater and recommendations for agriculture. https://www.oecd.org/greengrowth/sustainable-agriculture/Challenges%20of%20groundwater%20use.pdf.  The findings presented are already evident as significant, this resource would situate how significant for any reader.

The following comments are questions, suggestions and required changes. The write up as is keeps it understated and straight forward without many supporting citations of, for example, how significant and common overexploitation of groundwater/aquafers is and how important it is to resolve. This is a style of writing that might be preferred by the authors, and is acceptable. Some suggestions encourage citation or support from international examples, which can be uptaken or not, depending on your preference for presenting your findings.

Some questions indicate a need for clarity which could be presented in a discussion section, even a recommendations section. With significant specific findings for each subsection summarised.

Specific comments:

Abstract:

The abstract can be slightly better written however line 18-24 should be kept. The amount stated as increased 40% is not specified in findings only in abstract and in conclusion. Line 17 has a full stop that isn’t needed, and the characteristic of being a large scale export state or region could be included here.

Table 3.

Specify which of these agricultural water use programs address groundwater depletion and how they do. Two are identified in conclusion.

Line 295-297: Significant finding, it could be more obviously stated as significant with supportive international findings as to why groundwater depletion and overexploited aquifers are a significant finding for response in agricultural areas.

Line 329-330: Is the change in agricultural systems as more intensive over reduced land area with any stated intention to improve water use, according to any of the water programs identified in table 3, assuming it is part of irrigation systems listed.

Line 335-337: If it is inside the scope of your article you could consider whether intensive agricultural systems by area ha, despite decreased land use for agriculture are part of the reason for overuse of water.

Line 348-349: What percentage are rained agriculture of the ha land used for agriculture? Is it another significant factor for groundwater depletion, or overexploitation.

Line 352-355: Could write up as a discussion or recommendations section, or significant findings listed from results. As most the emphasised concluded points are about commercial agriculture and irrigated systems.

Line 358-359, expenditure does not lead to higher productivity in the sector, it is not dependent on government incentives. Significant finding could be in conclusion.

6. Conclusion

Line 364-365: almost indicates an allocation to compensate damage caused by the technification of irrigation systems. 

While these conclusion adequately state the most significant of findings with significant recommendations they could connect to existing practice to recommendation. Irrigation programs and intensified by area agriculture, could be improved by MRP? (Should be MPI?) And Better water use in the field, to prioritise water recharge, as 50% in the study area is irrigated. As an adapted rather than extremely different practice or encouraged program for the region, and specific recommendations for improvement.  

Line 378: Are there other management approaches that are not specific to water management and that have indirect impact on irrigation and water use for agriculture? Could they be included in recommendations.

Is the decrease in agricultural activities by land use/ha because of reduced availability of groundwater, or only because of intensive land use.

Line 380: With the fact that these are areas of large scale export, is there scope for recommendations regarding sourcing practices, as well as investment guidelines. Or would these guidelines regulate sourcing for export practices. (Point might be outside of scope of the article, as specific to government investment)

This is a significant specific finding for the researched community, region which could be supported as to significance by citation.

An advanced consideration from what is presented here is any government policy for ‘reducing water deficit’, or ‘groundwater recharge’. If part of the water management programs listed in table 3, or specific policies, or as regulation for export.

Comments on the Quality of English Language

The English expression does not require much editing.

Author Response

Perspective of water use programs in agriculture in Guanajuato

Response to Reviewer 2 Comments

1. Summary

2. Questions for General Evaluation

Reviewer’s Evaluation

Response and Revisions

Does the introduction provide sufficient background and include all relevant references?

Not applicable

Not applicable

Are all the cited references relevant to the research?

Not applicable

Not applicable

Is the research design appropriate?

Not applicable

Not applicable

Are the methods adequately described?

Not applicable

Not applicable

Are the results clearly presented?

Not applicable

Not applicable

Are the conclusions supported by the results?

Not applicable

Not applicable

3. Point-by-point response to Comments and Suggestions for Authors

Comments 1: Specify which of these agricultural waters use programs address groundwater depletion and how they do. Two are identified in conclusion.

Response 1:

Comments 2: Line 295-297: Significant finding, it could be more obviously stated as significant with supportive international findings as to why groundwater depletion and overexploited aquifers are a significant finding for response in agricultural areas..

Response 2: we are agreed, we added the data on other countries and it moves to line 370 to 379

Comments 3: Line 329-330: Is the change in agricultural systems as more intensive over reduced land area with any stated intention to improve water use, according to any of the water programs identified in table 3, assuming it is part of irrigation systems listed

 Response 3: no it doesn’t

 Move to line 430, The change in agricultural system is just one factor that can be improve the water management, but implies technology, infrastructure and investment that cannot be available for all

Comments 4: Line 335-337: If it is inside the scope of your article you could consider whether intensive agricultural systems by area ha, despite decreased land use for agriculture are part of the reason for overuse of water

Response 3: is not the scope, but we value your observation and add some changes and bibliographies move to line 436 to 444

Comments 5: Line 348-349: What percentage are rained agriculture of the ha land used for agriculture? Is it another significant factor for groundwater depletion, or overexploitation.

Response 5:  we value your observation, we add sone observations  in line 447-460

Comments 6: Line 352-355: Could write up as a discussion or recommendations section, or significant findings listed from results. As most the emphasized concluded points are about commercial agriculture and irrigated systems.

Response 6: we agreed the conclusion was modified

Comments 7 Line 358-359, expenditure does not lead to higher productivity in the sector, it is not dependent on government incentives. Significant finding could be in conclusion.

Response 7: we agreed it was added to a conclusion in line 530 to 532

Comments 8: Conclusion Line 364-365: almost indicates an allocation to compensate damage caused by the technification of irrigation systems While these conclusion adequately state the most significant of findings with significant recommendations they could connect to existing practice to recommendation. Irrigation programs and intensified by area agriculture, could be improved by MRP? (Should be MPI?) And Better water use in the field, to prioritise water recharge, as 50% in the study area is irrigated. As an adapted rather than extremely different practice or encouraged program for the region, and specific recommendations for improvement.  

Response 8: we value your observation, we add some suggest in line 545-547

Comments 9: Line 378: Are there other management approaches that are not specific to water management and that have indirect impact on irrigation and water use for agriculture? Could they be included in recommendations the decrease in agricultural activities by land use/ha because of reduced availability of groundwater, or only because of intensive land use

Response 9: we value your observation; we add some sugest in line 554-557

Comments 10: Line 380: With the fact that these are areas of large scale export, is there scope for recommendations regarding sourcing practices, as well as investment guidelines. Or would these guidelines regulate sourcing for export practices. (Point might be outside of scope of the article, as specific to government investment)

Response 10: we value your observation; but is not the scope

Comments 11: This is a significant specific finding for the researched community, region which could be supported as to significance by citation.

Response 11: we value your observation; we added citation of articles related of the region

Comments 12: An advanced consideration from what is presented here is any government policy for ‘reducing water deficit’, or ‘groundwater recharge’. If part of the water management programs listed in table 3, or specific policies, or as regulation for export.

Response 11: we appreciated your observation

5. Additional clarifications

Reviewer 3 Report

Comments and Suggestions for Authors

The article entitled Perspective of water use programs in agriculture in Guanajuato approaches a topical current issue in the studied geographic context.

The paper has a good logic flow and its results, mainly descriptive, are well argumented.

The major issue in this article is the methodology which may be accepted for a scientific repport but is not enough innovative and/or analytic for a scientific paper.

In my opinion the main issues to be improved by the authors of the paper would be therefore:

1. To point out the aim / purpose and and its scientific objectives in the given context (this should be done in both abstract and at the end of introduction)

2. Introduction should also clearly emphasize the need and usefulness of such a study. What is its utility and for whom ?

3. Please revise methodology – First, in the 2.1 Study area subchapter provide a map of the studied area.

I personally agree with documentation as a first methodologic level of the study and if considered part of the methodology this should be explained also through the difficulty of the endeavour...in the given context, as necessity ....as usefulness of this study (Are these data available...? to what scientific purpose they served....?)

However the main problem, as also explained above, is that documentation alone could not support currently the publication of a paper in a Q1 journal. As the topic is interesting and important in the context and well argumented by the authors I would kindly suggest to complement their results and analysis through either quantitative analysis (e.g. maybe regressions...as models would ask for more variables) OR qualitative (interview with one / two major key stakeholders on the issue). This could complement results subchapters and make a more consistent analysis from the scientific point of view.

4. Methodology should also be mentioned in the abstract.

5. After completing methodology and results accordingly ...Discussion and conclusion should stick around the research objectives and conclude the paper according to its main results.

6. At the end of conclusions it is also important to state again the gap filled by the study (research, theoretical, practical...etc.) and also its limits.

Please take into consideration all the above for further improvements of the paper. All these comments were made to help the authors publish the results of their work in Agriculture mdpi journal.

Author Response

Comments 1: [To point out the aim / purpose and and its scientific objectives in the given context (this should be done in both abstract and at the end of introduction) ].

Response 1: Agree.  The objective was placed at the end of the introduction and in the summary

Comments 2: [2. Introduction should also clearly emphasize the need and usefulness of such a study. What is its utility and for whom ?].

Response 2: Agree. Added information from lines 32 to 45

Comments 3: [Please revise methodology – First, in the 2.1 Study area subchapter provide a map of the studied area. I personally agree with documentation as a first methodologic level of the study and if considered part of the methodology this should be explained also through the difficulty of the endeavour...in the given context, as necessity ....as usefulness of this study (Are these data available...? to what scientific purpose they served....?) However the main problem, as also explained above, is that documentation alone could not support currently the publication of a paper in a Q1 journal. As the topic is interesting and important in the context and well argumented by the authors I would kindly suggest to complement their results and analysis through either quantitative analysis (e.g. maybe regressions...as models would ask for more variables) OR qualitative (interview with one / two major key stakeholders on the issue). This could complement results subchapters and make a more consistent analysis from the scientific point of view].

Response 3: Agree. We Added a map of the study area where the state of Guanajuato is located, as well as the geographical limitation of the aquifers on page 4 line 128. In relation to the suggestion of adding qualitative information by applying interviews, given the human and economic resources required, it is not possible to develop it, which is why it was obtained by checking if the information has a normal distribution and carrying out linear regression models. This information was added in methodology on page 5 line a151, the results of this analysis that focuses on „the multiple linear regression model, where the average annual availability of groundwater (DMA) is predicted based on the variables year, budget allocated to public programmes related to the use and utilisation of water in the agricultural sector, and the agricultural land area dedicated to irrigated, rain-fed, and protected agricultura” se encuentra en la pagina 11 linea 431.

Comments 4: [Methodology should also be mentioned in the abstract]

Response 4: Agree. We  Added in the Abstract in line 18.

Comments 5: [After completing methodology and results accordingly ...Discussion and conclusion should stick around the research objectives and conclude the paper according to its main result].

Response 5: Agree. Once the objective was added and the methodology was complemented, the discussion was expanded and the conclusion clarified, you can see the text highlighted in red in lines 332 to 342, 352 to 361; 417 to 430.

Comments 6: At the end of conclusions it is also important to state again the gap filled by the study (research, theoretical, practical...etc.) and also its limits.

Response 6: AgreeIt was structured in its last paragraph to take into consideration the general objective. You can see the text highlighted in red on page 13.

Reviewer 4 Report

Comments and Suggestions for Authors

The paper needs comprehensive editing and revision. 

The abstract needs to be revised according to the findings based on your analysis.

The manuscript's content is not organized and elaborated well, and the facts justifying the study's objectives are not focused enough. I recommend the authors revise the manuscript comprehensively. 

Comments on the Quality of English Language

 Extensive English editing is required. 

Please refer to each paragraph; some paragraphs contain only a single sentence. Usually, a paragraph needs three sentences, and there should be a link between paragraphs to have a proper flow. 

Author Response

Comments 1: The paper needs comprehensive editing and revision. 

Response 1: agree, we have a comprehensive language revision

Comments 2: The abstract needs to be revised according to the findings based on your analysis.

Response 3: agree we are modified the abstract

Comments 4: The manuscript's content is not organized and elaborated well, and the facts justifying the study's objectives are not focused enough. I recommend the authors revise the manuscript comprehensively. 

Response 3: agree, we have modifications thought the document

Comments 5:  Extensive English editing is required. 

Response 5:  agree the document was revised in terms of English editing

Comments 6: Please refer to each paragraph; some paragraphs contain only a single sentence. Usually, a paragraph needs three sentences, and there should be a link between paragraphs to have a proper flow. 

Response 6: agree, all the paragraphs are well revised.

Round 2

Reviewer 1 Report

Comments and Suggestions for Authors

In the revised manuscript, you have provided a detailed response to my review comments and improved the manuscript.

Comments on the Quality of English Language

Minor editing of English language required

Author Response

We greatly appreciate your comments; the English grammar has been reviewed again

Reviewer 3 Report

Comments and Suggestions for Authors

The second version of the paper entitled Perspective of water use programs in agriculture in Guanajuato is a much improved version of the initial manuscript. Certain aspects would need however a more in depth approach and I recommend the authors to still reflect on below aspects in view of publication of their research:

- Beside the aim or the general purpose the paper the authors are recommended to define also, as already pointed out in previous report, the scientific objectives (or research questions or the hypotheses of the study). They should be further directly connected to results and further debated in the discussion and/or conclusion chapter which is too short and does not exploit enough the implications of research findings...theoretical and practical implications of this study should be pointed out...who may be interested in such a modelling equation and why...

- In the end the paper should explicitly underscore for the readers which is the perspecive of water use programs in agriculture in this Mexican state as emphasized by the quantitative models... Is this similar for other states.... is there a particular situation for this one... ? Please extend conclusions and debate or discuss the findings related to the actual context...to external or internal factors that may further interfere in this equation and which maintain or not in the future the validity of these scientific results. These are extensive and conssistent addings that would problematize theory and emphasize practical implications of your method as related to the context. My suggestion is to make a combined chapter Discussion and conclussion and to add some consistent paragraphs to it. Please feel free to consider my content suggestions or other aspects that you find useful as experts in the studied context.

- At the end of conclusion, as also required in previous report, the study should also express its limits and possible future research orientations. Documentation and statistic methods offer limited quantitative inputs for such complex issues, involving different types of users / farmers (are there differences in the water usage...great / small exploitations...?) and governmental stakeholders that involve also complex qualitative and policy perspectives in this relation. This was also pointed out along suggestions and comments in previous review report and even if not part of the present research, the qualitative input could not be skipped by the study and may be underlined as a complementary missing / future perspective.

Author Response

For research article

Perspective of water use programs in agriculture in Guanajuato

Response to Reviewer 3 Comments

1. Summary

2. Questions for General Evaluation

Reviewer’s Evaluation

Response and Revisions

Does the introduction provide sufficient background and include all relevant references?

Not applicable

Not applicable

Are all the cited references relevant to the research?

Not applicable

Not applicable

Is the research design appropriate?

Not applicable

Not applicable

Are the methods adequately described?

Not applicable

Not applicable

Are the results clearly presented?

Not applicable

Not applicable

Are the conclusions supported by the results?

Not applicable

Not applicable

3. Point-by-point response to Comments and Suggestions for Authors

Comments 1: Beside the aim or the general purpose the paper the authors are recommended to define also, as already pointed out in previous report, the scientific objectives (or research questions or the hypotheses of the study). They should be further directly connected to results and further debated in the discussion and/or conclusion chapter which is too short and does not exploit enough the implications of research findings...theoretical and practical implications of this study should be pointed out...who may be interested in such a modelling equation and why...

Response 1: we agreed your observation and corrected the connected results and findings

Comments 2: in the end the paper should explicitly underscore for the readers which is the perspecive of water use programs in agriculture in this Mexican state as emphasized by the quantitative models... Is this similar for other states.... is there a particular situation for this one... ? Please extend conclusions and debate or discuss the findings related to the actual context...to external or internal factors that may further interfere in this equation and which maintain or not in the future the validity of these scientific results. These are extensive and conssistent addings that would problematize theory and emphasize practical implications of your method as related to the context. My suggestion is to make a combined chapter Discussion and conclussion and to add some consistent paragraphs to it. Please feel free to consider my content suggestions or other aspects that you find useful as experts in the studied context.

Response 2: we extremally value your suggestion, we agreed not to combine discussion and conclusions but also we agreed to add some one of you very acure observation a add international and national comparison citation

Comments 3: - At the end of conclusion, as also required in previous report, the study should also express its limits and possible future research orientations. Documentation and statistic methods offer limited quantitative inputs for such complex issues, This was also pointed out along suggestions and comments in previous review report and even if not part of the present research, the qualitative input could not be skipped by the study and may be underlined as a complementary missing / future perspective.

 Response 3: we agreed and add the conclusion to a more accurate

5. Additional clarifications

Reviewer 4 Report

Comments and Suggestions for Authors

The authors have extensively improved the manuscript, which includes valuable information for a general readership. However, extensive English editing through an editorial service is required before publishing it. 

Comments on the Quality of English Language

Extensive English editing through an editorial service is required before publishing it. 

Author Response

For research article

Perspective of water use programs in agriculture in Guanajuato

Response to Reviewer 3 Comments

1. Summary

2. Questions for General Evaluation

Reviewer’s Evaluation

Response and Revisions

Does the introduction provide sufficient background and include all relevant references?

Not applicable

Not applicable

Are all the cited references relevant to the research?

Not applicable

Not applicable

Is the research design appropriate?

Not applicable

Not applicable

Are the methods adequately described?

Not applicable

Not applicable

Are the results clearly presented?

Not applicable

Not applicable

Are the conclusions supported by the results?

Not applicable

Not applicable

3. Point-by-point response to Comments and Suggestions for Authors

Comments 1: The authors have extensively improved the manuscript, which includes valuable information for a general readership. However, extensive English editing through an editorial service is required before publishing it. 

Response 1: we agreed your observation and revised the English language

5. Additional clarifications
